# CHSI costing study–Challenges and solutions for cost data collection in private hospitals in India

**Maninder Pal Singh[1,2], Riya Popli[1], Sehr Brar[1], Kavitha Rajsekar[3], Oshima Sachin[3], Jyotsna Naik[3], Sanjay Kumar[4], Setu Sinha[4], Varsha Singh[4], Prakash Patel[5], Ramesh Verma[6], Avijit Hazra[7], Raghunath Misra[7], Divya Mehrotra[8], Sashi Bhusan Biswal[9], Ankita Panigrahy[9], Kusum Lata Gaur[10], Jai Prakash Pankaj[10], Dharmesh Kumar Sharma[10], Kondeti Madhavi[11], Pulaganti Madhusudana[11], K. Narayanasamy[12], A. Chitra[12], Gajanan D. Velhal[13], Amit S. Bhondve[13], Rakesh Bahl[14], Sharminder Kaur[14], Shankar Prinja[1,15]***

**1** Department of Community Medicine & School of Public Health, Postgraduate Institute of Medical Education & Research, Chandigarh, India, **2** Department of Global Health & Development, London School of Hygiene & Tropical Medicine, London, United Kingdom, **3** Department of Health Research, Ministry of Health and Family Welfare, Government of India, New Delhi, India, **4** Indira Gandhi Institute of Medical Science, Patna, Bihar, India, **5** Surat Municipal Institute of Medical Education & Research, Surat, Gujarat, India, **6** Pt. B.D.Sharma Post Graduate Institute of Medical Sciences, Rohtak, Haryana, India, **7** Institute of Postgraduate Medical Education & Research, Kolkata, West Bengal, India, **8** King George's Medical University, Lucknow, Uttar Pradesh, India, **9** Veer Surendra Sai Institute of Medical Sciences and Research, Burla, Odisha, India, **10** Sawai Man Singh Medical College, Jaipur, Rajasthan, India, **11** Sri Venkateswara Medical College, Tirupati, Andhra Pradesh, India, **12** Madras Medical College, Chennai, Tamil Nadu, India, **13** Seth G S Medical College & KEM Hospital, Mumbai, Maharashtra, India, **14** Government Medical College, Jammu, Jammu & Kashmir, India, **15** National Health Authority, Government of India, New Delhi, India

* shankarprinja@gmail.com

**Data Availability Statement:** All relevant data are within the manuscript and its Supporting Information files.

## Abstract

### Introduction

*Ayushman Bharat Pradhan Mantri Jan Aarogya Yojana* (AB PM-JAY) has enabled the Government of India to become a strategic purchaser of health care services from private providers. To generate base cost evidence for evidence-based policymaking the Costing of Health Services in India (CHSI) study was commissioned in 2018 for the price setting of health benefit packages. This paper reports the findings of a process evaluation of the cost data collection in the private hospitals.

### Methods

The process evaluation of health system costing in private hospitals was an exploratory survey with mixed methods (quantitative and qualitative). We used three approaches–an online survey using a semi-structured questionnaire, in-depth interviews, and a review of monitoring data. The process of data collection was assessed in terms of time taken for different aspects, resources used, level and nature of difficulty encountered, challenges and solutions.

### Results

The mean time taken for data collection in a private hospital was 9.31 (± 1.0) person months including time for obtaining permissions, actual data collection and entry, and addressing

**Funding:** The study is funded by the Department of Health Research (DHR), Government of India.

**Competing interests:** The authors have declared that no competing interests exist.

queries for data completeness and quality. The longest time was taken to collect data on human resources (30%), while it took the least time for collecting information on building and space (5%). On a scale of 1 (lowest) to 10 (highest) difficulty levels, the data on human resources was the most difficult to collect. This included data on salaries (8), time allocation (5.5) and leaves (5).

## Discussion

Cost data from private hospitals is crucial for mixed health systems. Developing formal mechanisms of cost accounting data and data sharing as pre-requisites for empanelment under a national insurance scheme can significantly ease the process of cost data collection.

## Introduction

In India, the government launched the flagship program, *Ayushman Bharat* (AB) in 2018 to provide universal health coverage (UHC). It is a two-pronged strategy providing both primary care through Health & Wellness Centres (AB HWCs), and secondary and tertiary care services through the *Pradhan Mantri Jan Arogya Yojana* (AB PM-JAY) [1].

AB PM-JAY is a tax-funded national health insurance scheme, for providing access to hospitalisation services provided through a network of empanelled public and private hospitals with coverage of ₹ 500,000 (≈US$ 7000) per family per year for 100 million families [1]. This provides the government with an opportunity to become a strategic purchaser by paying the empanelled health care providers (EHCPs) for a set of pre-defined health benefit packages (HBPs). Strategic purchasing is one of the basic principles as part of the health financing reforms to achieve universal health coverage (UHC) by equitable and efficient distribution of resources to achieve maximum gains [2, 3]. Further, the Government of India established Health Technology Assessment India (HTAIn) Board, a health technology assessment agency with a mandate to support evidence-based resource allocation, by providing evidence on the cost-effectiveness of interventions and services [4].

For strategic purchasing in AB PM-JAY, robust evidence on health care cost is crucial for pricing decisions. However, there is a paucity of evidence on health system costs due to weak data management systems and limited electronic health records [5, 6]. Moreover, the published studies and national cost database are either focussed on primary care costs or are limited to selected surgical procedures, disease conditions and public sector hospitals [5, 7–14]. On the other hand, AB-PMJAY focusses on the provision of tertiary as well as secondary care, with significant contribution from the private sector [15, 16]. However, evidence on the health system cost data from private hospitals is available from few small scale studies which do not represent the heterogeneity within the private hospitals [9, 17].

Private sector hospitals in India have grown and diversified in terms of service provisioning, infrastructure, ownership to name a few [18]. There is no centralized registration of private hospitals in India. Private sector comprises for-profit and not for-profit hospitals ranging from large corporate hospitals, super-speciality facilities to medium and small nursing homes and even single doctor dispensaries [19]. In India, there are 43,486 private hospitals accounting for 62% of the health infrastructure of the country [20].

The health system costing studies require a complex set of information for different input resources which is a labour intensive and time consuming. The process on data collection is contingent upon the costing approach (financial or economic), perspective (health system or patient), methods (top-down or bottom-up or mixed), willingness to share data, availability of

data management systems to name a few [21, 22]. These challenges are further exacerbated in the low-and middle-income countries (LMICs) such as India due to insufficient data management systems and requirement of data extraction from physical registers [23–25].

To generate credible evidence on the cost of healthcare services using a standardized costing methodology and representative of both public and private hospitals, the Costing of Health Services in India (CHSI) study was commissioned in 2018 [17]. The objective of the CHSI study is to estimate the unit cost of individual healthcare services and the health benefit packages (HBPs) covered under the AB PM-JAY.

The CHSI study is a four-phased first national costing study. Phase I covered public sector tertiary care hospitals and its estimates were used in the price revision of AB PM-JAY in 2019 [26]. A process evaluation of cost data collection from public sector tertiary care hospitals was undertaken highlighting key challenges in data collection and potential way forward [24]. However, private sector hospitals are much more diverse in terms of size, ownership, structure, and geographical distribution to name a few [18]. Further, the nature of information systems, level of transparency, and willingness to share data is different in each private hospital. Therefore, considering the heterogeneity and predominant role in AB PM-JAY it's important to explore the process of cost data collection in private hospitals, and its related challenges. This paper reports the findings of the process evaluation of the cost data collection in the private hospitals as part of the CHSI phase III. It outlines the process of data collection, challenges, and way forward from the first nationally representative costing study of private hospitals of India.

## Materials and methods

### Costing of Health Services in India (CHSI) study

The objective of the CHSI study was to provide cost evidence for pricing decisions in AB PM-JAY HBPs and HTA studies. The CHSI phase I covered 8 specialities from 11 public sector tertiary care hospitals across 11 Indian states. Phase II of the CHSI study covered 27 public sector district hospitals providing secondary health care from 9 states. In phase III of the CHSI study, 38 private hospitals were sampled from 10 states. The sampling details are provided in S1 Table. A stratified sampling methodology was used for selection of private hospitals, wherein 75% of the sampled hospitals were for-profit and 25% were not for-profit hospitals. This was based on the proportion of private hospitals in each of the category [18]. Further, the location of hospital (i.e. classification of cities used by the government in India tier 1, tier 2 and tier 3) and number of hospitalisation beds (up to 50 bedded, 50–200 bedded and more than 200 bedded) was used as a sampling criteria. No single speciality hospital was included in the study. From each state a sample of 3–4 hospitals was included in the study.

The study uses standard principles of economic costing and the micro-costing approach [17, 27]. A mixed micro-costing approach (top-down and bottom-up) from a payer's perspective was used to estimate health system cost. While a pure bottom-up costing methodology may provide the most detailed and granular estimation of cost, lack of disaggregated data and electronic data management systems makes its execution difficult for certain resources. Hence, for resources such as utilities, maintenance, and in certain cases the drugs and consumables, The top-down methodology was used. However, our top-down method provided much more granular level of aggregation at the level of a cost centre, rather than entire health facility [28]. Such a methodology is commonly applied in the costing studies carried out in India and other LMIC settings [17, 29]. The next step was identification of cost centres such as outpatient (OP) department, inpatient (IP) department, operation theatre (OT), intensive care unit (ICU) to name a few [17]. It was followed by identification and measurement of outputs and inputs

used by each cost centre to calculate the unit cost of each output generated or service delivered like cost per OP visit, cost per bed-day in IP or ICU, cost per surgery in OT. Within each cost centre human resources accounted for highest share in the cost of service delivery. As, human resources are shared between different cost centres, time allocation interviews were conducted to understand the work pattern. The time allocation interviews were undertaken for each category of the staff such as specialist, doctors, nurses, technicians to name a few. Within each staff category at least 25% of the staff was interviewed [17]. The time allocation data collection tool in S2 Table.

**Data collection.** The process evaluation of health system costing in private hospitals employed three approaches, namely an online survey using a semi-structured questionnaire, in-depth interviews and the review of monitoring data. The data collection tool has been provided in S3 Table. Data was collected from the study teams in terms of time taken, resources used, challenges faced and solutions attempted in collecting cost data in private sector facilities. The study tools were adapted from the process evaluation study undertaken in public sector tertiary hospitals from CHSI phase I [24].

*Online survey*. For the online survey, a semi-structured tool was designed to collect information on processes of data collection and the challenges involved in primary data collection for health system costing. It had both quantitative and qualitative parameters divided into five sections. In the first section, information about the field study team such as location, service capacity, nature and type of private hospitals and departments where data was collected, the process of permissions, staff recruitment and training were collected. The second section contained both objective and subjective questions to understand the process of obtaining permissions and critical challenging areas respectively. The third section was used to collect detailed information on processes adopted, as well as the nature and level of difficulties in collecting data on quantity and prices for each input resource such as human resources, capital, drugs, consumables, overheads, and data on service volume. This section was further divided into three sub-sections. The first sub-section had multiple-choice questions on the process of data collection. The second sub-section was a tabulation for information on time spent on each aspect of data collection, sources of data accessed, challenges, and solutions implemented to overcome such problems. At last, for each input resource, the respondent had to rate the level of difficulty in cost data collection on a Likert scale (1 to 10), where no difficulty in data collection means 1 and 10 implied data collection was not possible. The third sub-section focussed on additional data required for the costing of surgical procedures. The last two sections focused on training and supervision and inputs to improve ease of data collection in private hospitals. The study participants included field study teams of 7 states (Andhra Pradesh, Bihar, Gujarat, Odisha, Rajasthan, Uttar Pradesh, and West Bengal) involved in the CHSI study for data collection from 21 private hospitals.

*In-depth interviews*. Virtual in-depth interviews were conducted after the online survey was submitted. It was an open-ended interview with the data collection teams and central team to obtain detailed information on the key identified areas. The main areas of discussion were the requirement of permissions, physical records, identification of data sources and learnings to evolve the data collection tools specifically for health system costing in private hospitals. Further, how learnings and experiences from public sector tertiary and district hospitals costing were helpful were also discussed.

*Review of monitoring data and interviews with central team*. The CHSI central data analysis team monitors the state-level data collection and maintains the data on time taken to complete data collection, need for on-the-job support in collecting data, quality of data etc. This data maintained by the central analysis team was used to calculate the time spent on data collection

by each field study team. The central team was interviewed to understand the process of quality assurance.

A written informed consent was taken from all the study participants involved in online survey, in-depth interviews, and interviews with central team. Ethical approval for the CHSI study was obtained from the Institutional Ethics Committee & Institutional Collaborative Committee of Post Graduate Institute of Medical Education and Research Chandigarh.

## Data analysis

**Quantitative data.**   The primary data collected through the online survey tool was exported and analysed using MS Excel 2019. Continuous variables were summarised as mean, standard error, range, median, and interquartile range. Categorical variables were analysed through frequency or percentages. The total time for data collection was calculated in two ways, first the total time taken in the process of data collection and last the person-days were calculated to derive actual time spent on data collection after excluding the negotiating and waiting time period. A comparison of time spent on each input resource was undertaken with its share in the total cost.

**Qualitative data.**   The qualitative data was obtained from an online survey and in-depth interviews. The in-depth interviews focussed the discussion on key challenges and best practices in data collection. The focus of the qualitative interviews of the central team was to derive information on quality assurance. The subjective answers as part of the online tool were analysed by identifying common thematic areas across different field study teams.

## Results

### Staff profile and training

The data on staff profile and training was derived from the online survey and review of monitoring data and interviews with central team. Each state data collection team had 3 project staff (2 field officers and 1 administration assistant) guided by 1 to 3 state co-investigators. The field data collection teams comprising of 3 full-time staff which worked for 40 hours a week except for Sundays and public holidays of the respective state. From these 7 state teams, 56% of the state co-investigators and 14% of the project staff were trained in an in-person workshop by the central team respectively. The remaining staff was trained locally and guided virtually by the central team.

### Time period for primary data collection

The data on time period for primary data collection, hospitals approached and refusal rate was calculated from the online survey and review of monitoring data and interviews with central team. The primary data collection was undertaken in 21 private hospitals from 7 states of India (Andhra Pradesh, Bihar, Gujarat, Odisha, Rajasthan, Uttar Pradesh, and West Bengal). (S1 File) In each state on an average 6 hospitals were approached to participate in the CHSI study. A minimum of 3 hospitals were approached in Odisha and Gujarat, and there was zero refusal rate. In Uttar Pradesh 7 hospitals were approached and 5 (71%) refused to participate in the study. In states of Rajasthan and West Bengal 9 hospitals were approached and there was a refusal rate of 67% in both the states. In Andhra Pradesh 6 hospitals were approached and 3 (50%) refused to consent for the study. The overall refusal rate among private hospitals across the states was 45%. Out of sampled hospitals, 11 were less than 50 bedded hospitals and 8 were 50–200 bedded hospitals. Based on the ownership, 62% [13] of the sampled hospitals were for-profit and the remaining 38% were not-for-profit hospitals.

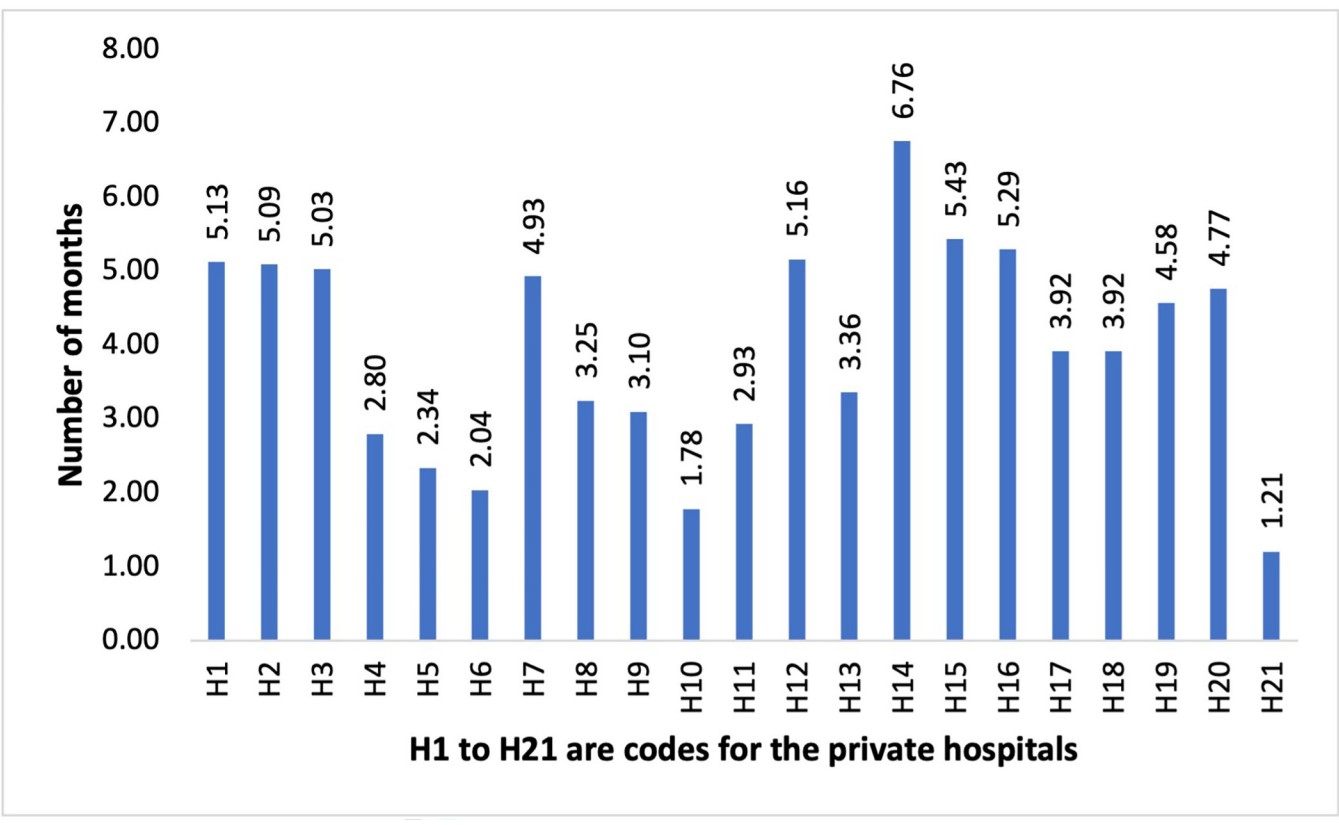

**Fig 1. Total time taken for data collection for each facility (in months).**

The mean time taken for overall data collection per private hospital for a field data collection team comprising of three full-time staff, which includes obtaining permissions, actual data collection and entry, queries and final submission, was 3.94 months respectively. The time of data collection ranged from 1.21 months to 6.76 months. (Fig 1) On an average, data collection in one hospital covering all its services required 9.31 (± 1.0) person months.

The highest proportion of time was spent on collecting data for human resources and their time allocation (28%), quantity and prices of equipment (22%), followed by 12% each on furniture, and drugs & consumables. Collecting data on capital/building consumed the least proportion of time (6%) (Fig 2).

For each input resource, the duration of data collection was stratified into time to obtain a relevant source of data and actual time (in person-days) spent on data collection. The highest time was spent to obtain the records for accessing data on staff salaries while it took the least time to determine rental price. The actual time of data collection (in person-days) was maximum for obtaining detailed information on time allocation of personnel (median time 9.5 person-days) and minimum for determination of rental price (median time 1 person day). (Table 1) The share of time required for each type of input resources was calculated from the online survey.

Further, a comparison was undertaken between the proportion of time taken for data collection of each input resource and their contribution to the total cost. It was observed that human resources contribute the maximum in total cost (38%), as well as in time taken for data collection (28%). The capital/building and consumables & drugs contribute 44% to the total cost but only 18% of the time was spent to collect data for these inputs. On the other hand,

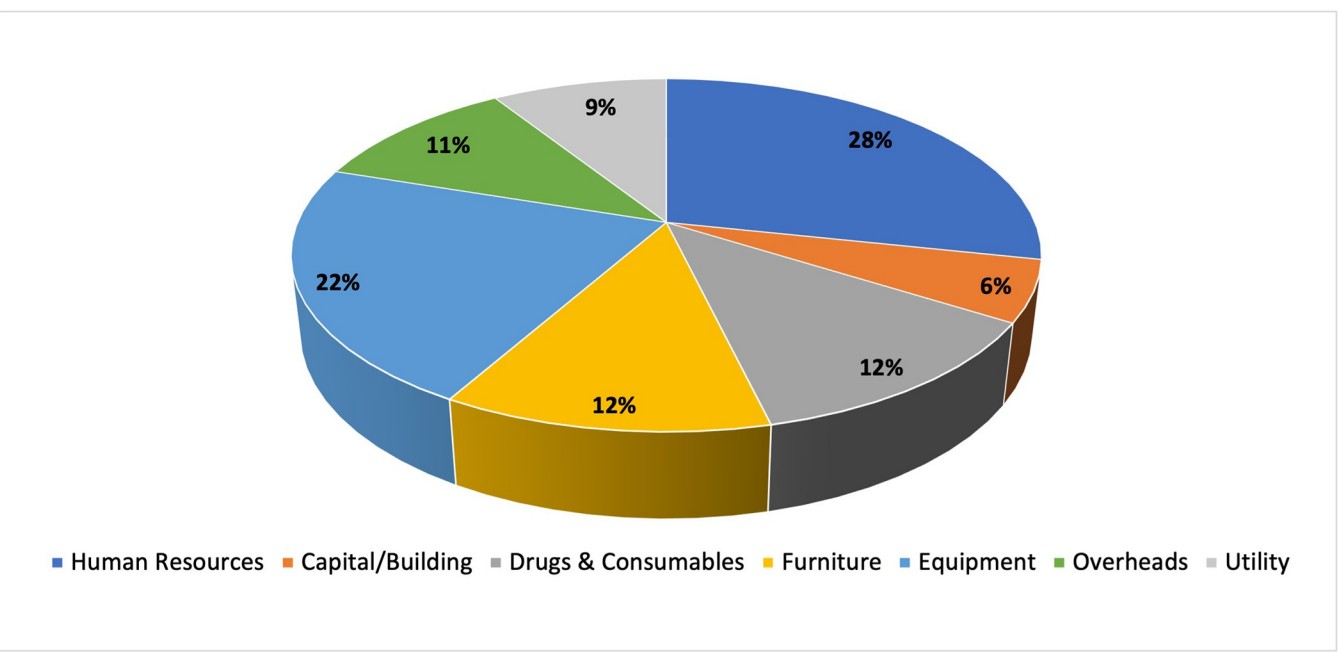

**Fig 2. Percentage share of time for each input resource.**

data on furniture and equipment consumed 34% of the time but contributes only 7% to the total cost (Fig 3).

### Processes of data collection for costing in private hospitals

The specific challenges and rating of difficulty in collecting data on each input resource are described in this section which are derived from the online survey and in-depth interviews with the state field teams. (Fig 4) The data relating to human resources i.e. salaries, leaves and time allocation interviews was most crucial and at the same time most difficult to collect. The data on salaries and leaves were collected from the accounts department. For time allocation, willingness and availability of the concerned staff were very important. Due to the busy schedule of medical personnel, the data collection teams had to make multiple rounds to complete the interviews. The involvement of senior staff members from the state teams to get interview time eases the process. The data on the space and building, as well as its rental price, were easier to collect. Obtaining data on the prices and utility of equipment in different procedures, was another aspect that received a high difficulty rating due to limited access in the operation theatres (OT). Further, the OT staff schedule is contingent upon the planned and emergency surgery which sometimes delayed the process of data collection.

The data on procurement prices for equipment, furniture, drugs and consumables were challenging as the private hospitals were not comfortable sharing it. The building maps were cross-checked physically to incorporate all the internal changes from the original plan. The rental price was determined based on a market survey of commercial prices in the vicinity of the private hospital.

An overview of data collection for different input resources was provided in Table 2. The data presented in the Table 2 was derived from the online survey and in-depth interviews.

### In-depth interview results

It was highlighted that approaching the management of the private hospital and having personal contacts with the management was quite useful. The signing of a non-disclosure

Table 1. Time required for cost data collection for each input resource.

| Input Source | Data type | Time required for collection | |
|---|---|---|---|
| | | Negotiation & waiting (Days) (Median, Interquartile Range) | Actual Collection (Person-days) (Median, Interquartile Range) |
| Human resource | HR Salary & Incentives | 12 (7.75–20) | 9 (4.5–24) |
| | Leave | 3 (2–4.5) | 3 (3–5.5) |
| | Time allocation | 11.5 (5.5–17) | 23 (9–45.0) |
| Physical area/ Building | Building area measurement | 3 (2–4.75) | 9.5 (4.25–11.5) |
| | Determination of rental price | 1 (1–1) | 1 (1–1) |
| Consumables /Drugs | Consumables used | 5.5 (2.25–15) | 8 (3–17) |
| | Prices of consumables | 5 (2.25–4.75) | 7.5 (3.75–9.25) |
| Furniture /Non-consumables | Furniture items used | 5.5 (2–7.75) | 9 (6.25–10) |
| | Prices of non-consumable items | 3.5 (2.25–5) | 5.5 (3.25–9) |
| | Information on average life of furniture items | 3 (2–4.75) | 3 (2–3) |
| Equipment | Equipment used | 5 (2.25–7) | 9 (5.25–11.5) |
| | Equipment procurement prices | 4.5 (2.25–5) | 10 (3.25–13.5) |
| | Average life of equipment | 3 (2.25–5) | 3.5 (2–10) |
| | Usage of equipment in different procedures | 4.5 (3–5) | 11 (6.25–13.5) |
| Overheads | Electricity | 2.5 (1–8.25) | 3.5 (2.25–5) |
| | Building Maintenance | 3 (1–8.5) | 3.5 (2.25–5) |
| | Equipment Maintenance | 3.5 (1–8.75) | 3.5 (2.25–5) |
| Utility | Laundry | 3 (1–4) | 4 (3–4) |
| | Dietetics | 5 (1.75–7.25) | 4 (2–5) |
| | Biomedical waste management | 3 (1–6.25) | 3 (2–3.75) |
| Service provision | Annual load data | 4.5 (2.5–5.75) | 6.5 (4.25–9.75) |

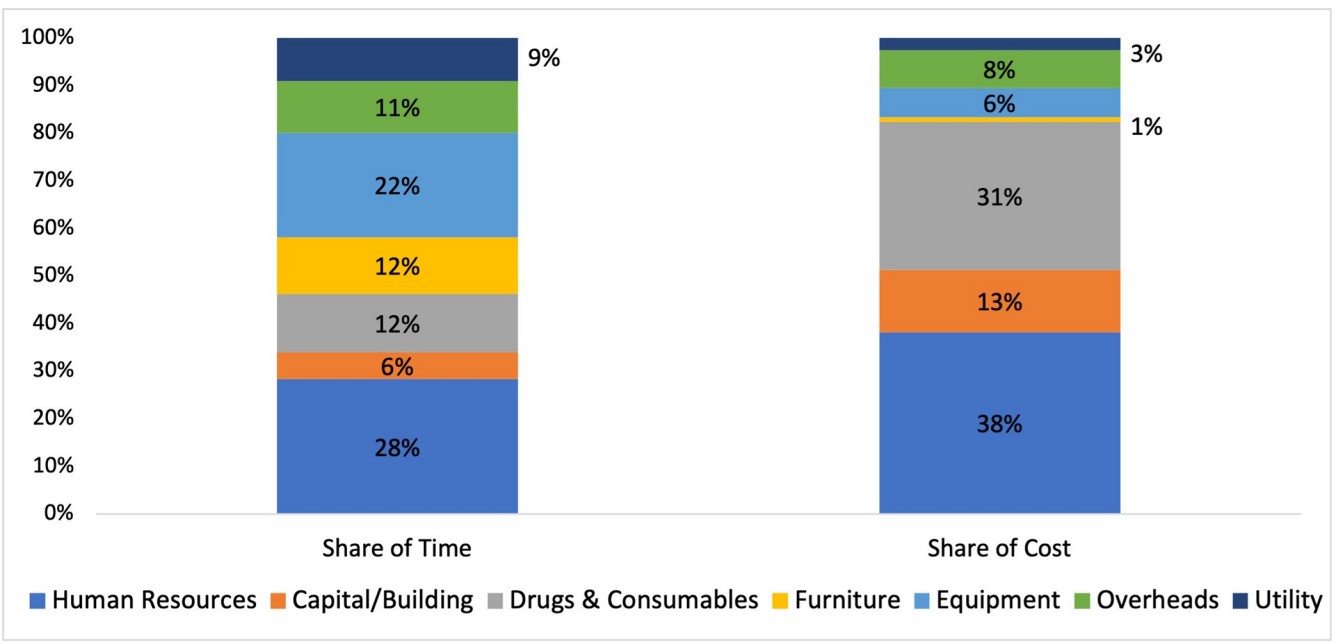

Fig 3. Percentage share of time and cost for each input resource.

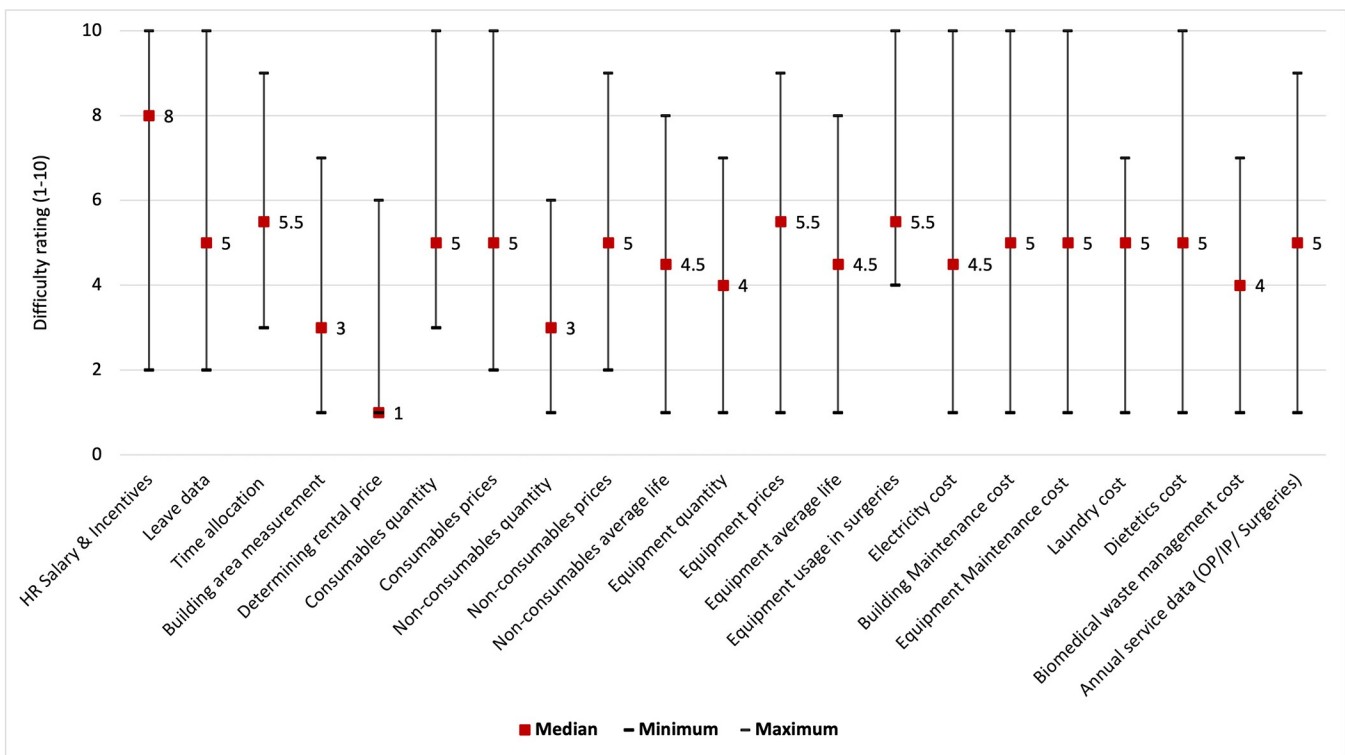

**Fig 4. Difficulty rating\* (1–10) for different input resources during cost data collection.**

agreement (NDA), assurance that the results generated will be anonymised and the name of the hospital will not be shared with any government body sets a positive tone with the management. Secondly, explaining the study methodology and data requirements is very crucial. Providing information on the disaggregation of data that is required for the costing, especially data on salaries and prices of equipment, furniture, drugs, and consumables, helped in data collection later. It should be clearly mentioned that data will only be used for the estimation of cost-of-service delivery and will not reflect negatively on the performance of an individual or hospital. Thirdly, a meeting with all the concerned persons such as hospital managers, administration staff, pharmacists etc from whom data will be collected was useful to sort the queries and obtain the cooperation of staff. Lastly, for some data such as salaries, patient load, length of stay and prices of items such as equipment, furniture, drugs, and consumables, it was reported that sharing the data entry tool with the hospital staff enabled them to fill the data directly, rather than being interviewed. The field staff reported not having been provided with direct access to source data in private hospitals.

## Discussion

In the PM-JAY scheme, 56% of the empanelled health care providers (EHCPs) are private hospitals. Private hospitals are catering to 63% of the PM-JAY claims volume, and 75% in terms of claims amount [16]. Given such a stake of participation of the private sector, it is important that the prices or provider payment rates need to be representative of the cost of providing care in private hospitals [16]. Currently, public and private hospitals are paid uniformly under the PMJAY scheme. Under the latest PM-JAY price revision in the 2022, the National Health Authority (NHA) introduced differential pricing based on the tier classification of the cities. This has made use of cost of services in both the public and the private hospitals [30].

**Table 2. Sources, challenges and potential solutions for cost data collection.**

| Data type | Level of data aggregation | Source and contact point | Common challenges | Potential suggestions |
|---|---|---|---|---|
| **Human resource data** | | | | |
| **HR Salary & Incentives** | Institution | 1. Hospital manager<br>2. HR head<br>3. Accountant | • Information on salaries is financially sensitive* | • Written or verbal permission from the owner or management of the hospital |
| **Leave** | Institution | 1. Hospital manager<br>2. Data operator/clerk<br>3. Accountant | • Data sharing hesitancy<br>• Lack of common records | |
| **Time allocation** | Individual | Personal Interview | • Busy working schedule<br>• Fear about negative reflection on performance and work | • Written or verbal permission from the owner or management of the hospital<br>• Initial interaction by senior staff members of data collection to solve the queries and doubts. |
| **Physical area/ building** | | | | |
| **Building area measurement** | Institution | 1. Hospital manager<br>2. Admin staff | • Variation in original maps and structural changes | • Use of smartphone applications for ease of work. |
| **Determination of rental price** | Institution | 1. Hospital manager<br>2. Pharmacist<br>3. Shopkeeper | • Difference in opinion on the rental price. | • Average price derived based on multiple information |
| **Consumables** | | | | |
| **Quantity of Consumables** | Institution or Department | 1. Pharmacist<br>2. Staff nurse in charge<br>3. Person in charge | • Lack of disaggregated data | • Expert inputs for disaggregation to cost centres |
| **Prices of consumables** | Institution | 1. Pharmacist<br>2. Admin staff<br>3. Person in charge | • Not be available for all items. | • Use multiple sources |
| **Equipment and non-consumables** | | | | |
| **Quantity** | Institution/ Department | 1. Hospital Manager<br>2. Admin staff<br>3. Person in charge | • Lack of disaggregated data | • Expert inputs for disaggregation to cost centres |
| **Prices of items** | Institution | 1. Hospital Manager<br>2. Admin staff<br>3. Person in charge | • Information on prices is financially sensitive*<br>• Missing information | • Written or verbal permission from the owner or management of the hospital |
| **Information on average life of items** | Department | 1. Staff nurse<br>2. Cost centre in charge<br>3. OT technician | • Missing information | • Use expert opinion from the personnel using the specific item |
| **Usage of equipment in different procedures** | Department/ Cost centre | 1. Doctors<br>2. Staff nurse<br>3. Cost centre in charge<br>4. OT technician | • Missing information | • Interview from the personnel using the specific items |
| **Overheads e.g., electricity, maintenance, laundry, dietetics, waste management** | | | | |

*(Continued)*

**Table 2.** (Continued)

| Data type | Level of data aggregation | Source and contact point | Common challenges | Potential suggestions |
|---|---|---|---|---|
| **Overhead costs for the service/ department** | Institution | 1. Hospital Manager 2. Admin staff 3. Person in charge | • Information on prices is financially sensitive* | • Written or verbal permission from the owner or management of the hospital |
| **Service provision data** | | | | |
| **OPD/IPD/ Surgeries** | Institution/ Cost centre | 1. Hospital Manager 2. Staff nurse in charge 3. Person in charge | • Operation theatre (OT) and inpatient (IP) records may not be disaggregated by procedure and speciality respectively. | • OT: Use surgical records • IP: Patient admission registers and expert opinion |

*The term financially sensitive means that the information pertaining to salaries and procurement prices are based on internal mechanisms and negotiations. There disclosure may influence the future negotiations for the private hospitals.

The CHSI study which was commissioned in 2018, aims to determine the cost of AB PM-JAY HBPs from tertiary and secondary public and private hospitals [26]. A process evaluation of study in public tertiary hospitals as part of phase I of the CHSI study was published earlier [24]. However, given the strong interest of costing in private sector hospitals, it is important to learn the processes, challenges and pragmatic solutions for efficient cost data collection in such private facilities.

In low-and middle-income countries (LMICs), cost data collection has been reported to have numerous challenges such as willingness to participate, sharing of data, the form of data management systems, lack of electronic records, lack of disaggregated data, multiple sources to name a few [21, 22, 24, 31]. Therefore, the present study was planned to understand the challenges and way forward for cost data collection in the private hospitals of India. The data on process evaluation was collected from 7 states where data was being collected from 21 private hospitals.

Overall, we found that the cost data collection in private hospitals was contingent upon the willingness of the hospitals to participate in the study and share their data. There was tendency in the state-level data collection teams to select the hospitals where they had personal contacts with the owner/s or management of the private hospital, it was easier to start the data collection. However, we adhered with the sampling framework. Initially, multiple hospitals were approached, and meetings were held but most of the private hospitals were hesitant to share the data.

The data collection in private hospitals pertaining to human resources in terms of salaries, leaves and time allocation was reported as most difficult and time-consuming by field investigating teams in terms of initially seeking permission, as well as actual data collection time. Based on our experience from CHSI study and previous costing studies, the study respondent (from private hospital) perceives the data collection may lead to some audit, which may reflect badly for them or institution. Moreover, since the prices for drugs, consumables and equipment has confidentiality issue related to the hospital profit margins. Similarly, the salary data has linked liability of taxes, which may not be revealed to avoid paying tax. Hence, there is reluctance to share the data, more commonly in private hospitals. In public hospitals, the reluctance to share data on quantity of resources consumed can only arise if the hospital records are not well maintained and there can be a mismatch between the hospital records and its reports submitted to higher facility. The data on time allocation may be influenced by the respondent's social desirability bias to be seen an performing in an ideal manner.

## Cost data collection: Public versus private hospitals

In comparison with the process of data collection in the public tertiary care hospitals and private hospitals, certain similarities and dissimilarities were highlighted. Online survey tool, in-depth interviews and review of monitoring data and interviews with central team was used to collect process evaluation data for both tertiary public and private hospitals [24]. For human resources salary and incentives, the median negotiation time is similar i.e. 12.5 days in public hospitals and 12 days in private hospitals. Further, the share of human resources in overall cost is around 40% in both public and private hospitals. The share of time spent on collecting data on equipment was also found to be the same (23%), as well as the rating of its difficulty.

The actual data collection time (person-days) for time allocation interviews is almost double in private hospitals as reported for public tertiary hospitals. The level of difficulty to access the data for equipment, furniture and consumables were higher in public hospitals as compared to the private hospitals. One of the reasons can be the difference in data management systems. In private hospitals, computer-generated reports of data were made available to the teams however in public tertiary hospitals data was extracted from physical registers. Another reason could be that multiple individuals were responsible for management within the same cost centre in the public sector hospitals, which increases the extent of permissions, need for explaining the purpose of data collection, and hence the overall data collection time. However, in the private hospital's data, extraction was dependent on quite a few staff members. The average time required for each public sector tertiary care hospital was 7 months, [24] but it was found to be 3.94 months for private hospitals. In the public sector tertiary care hospitals, data were collected from 3–4 departments/specialities with each department functioning as a separate unit thus requiring multiple permissions, negotiations, and interactions.

Further, the private sector hospital data collection was undertaken in phase III of the study. Prior to this, the state data collection teams had extensive experience in collecting data in the public sector tertiary and secondary care hospitals. Considering this, it is quite understandable that the teams took relatively lesser time in private hospitals to collect data.

## Limitations

Our study has a few limitations. Firstly, while the overall sampling plan was adhered to, there was a tendency of field data collection teams to choose the private hospitals where they had personal contacts with the owner/s or management of the hospital to facilitate data collection. Secondly, the Clinical Establishments (Registration and Regulation) Act (CEA) was enacted by the central government in 2010 [32]. The CEA is applicable to all systems of medicine and all the public and private hospitals. However, the implementation of CEA remains slow. Therefore, the reporting of mandatory cost data should be part of the empanelment requirement under the PM-JAY. Further, the implementation of CEA should be expedited and gradually reporting of cost of health services should be integrated to the provisions of the CEA. Thirdly, due to lack of centralized registration mechanisms for private hospitals and considering vast heterogeneity in organisation of private hospitals, the result of the present study may not be generalizable at the national level. However, the results will provide an importance guidance for the planning and execution of future rounds of cost data collection in the private hospitals.

## Roadmap for future costing studies in private hospitals

From a policy perspective to enable evidence-based price setting for AB PM-JAY by the National Health Authority (NHA), evidence on the cost of services from private hospitals is important information. While such costing studies would help set a baseline database, it would need to be regularly updated. However, considering the significant time and resources

required for such a detailed undertaking, it may be difficult to carry out studies in short intervals. A more sustainable way forward would be a system of regular reporting of cost data by the private hospitals to the NHA. This could be done in a standardized manner using a basic minimum set of information. Such a reporting of cost could become a pre-requisite for empanelment. Alternatively, the hospitals could be incentivized for the provision of this additional information. This will readily generate the cost evidence and at the same time provide an alternative for repeated primary data collection. Further, due consideration should be given for the data quality checks so that the reported cost data is not exaggerated or under-reported such as using several triangulation methods by comparing the data provided with similar unit prices obtained from other hospitals in the same city, or state, and with the price data from the National Health System Cost Database of India [33].

The diversity in the organisation of the private sector (ownership, services, geography, bed strength) complicates the designing, sampling, and data collection for a costing study. There should be an active engagement between the purchaser (government, NHA) and provider (private hospitals, hospital associations) to design a pre-defined format for regular reporting of cost data which protects the confidentiality of the providers and at the same time meets the standardized quality assurance measures. It will serve to create a synergy between the public and private sector for readily generating and sharing cost evidence and providing an alternative for repeated primary data collection during subsequent price revisions.

## Supporting information

**S1 Table. Costing of Health Services (CHSI) study sampling.**
(DOCX)

**S2 Table. Costing of Health Services (CHSI) time allocation tool.**
(DOCX)

**S3 Table. Private sector process evaluation survey for Costing of Health Services in India (CHSI) study.**
(DOCX)

**S1 File.**
(XLSX)

## Acknowledgments

We gratefully acknowledge the efforts of state data collection teams and staff of the private hospitals in sharing the information as part of the Costing of Health Services (CHSI) study.

## Author Contributions

**Conceptualization:** Maninder Pal Singh, Sehr Brar, Shankar Prinja.

**Data curation:** Maninder Pal Singh, Riya Popli, Sehr Brar, Sanjay Kumar, Setu Sinha, Varsha Singh, Prakash Patel, Avijit Hazra, Raghunath Misra, Divya Mehrotra, Sashi Bhusan Biswal, Ankita Panigrahy, Kusum Lata Gaur, Jai Prakash Pankaj, Dharmesh Kumar Sharma, Kondeti Madhavi, Pulaganti Madhusudana, K. Narayanasamy, A. Chitra, Gajanan D. Velhal, Amit S. Bhondve, Rakesh Bahl, Sharminder Kaur.

**Formal analysis:** Maninder Pal Singh, Riya Popli.

**Funding acquisition:** Shankar Prinja.

**Investigation:** Sehr Brar, Sanjay Kumar, Setu Sinha, Varsha Singh, Prakash Patel, Avijit Hazra, Raghunath Misra, Divya Mehrotra, Sashi Bhusan Biswal, Ankita Panigrahy, Kusum Lata Gaur, Jai Prakash Pankaj, Dharmesh Kumar Sharma, Kondeti Madhavi, Pulaganti Madhusudana, K. Narayanasamy, A. Chitra, Gajanan D. Velhal, Amit S. Bhondve, Rakesh Bahl, Sharminder Kaur.

**Methodology:** Maninder Pal Singh, Sehr Brar, Shankar Prinja.

**Project administration:** Maninder Pal Singh, Kavitha Rajsekar, Oshima Sachin, Jyotsna Naik, Sanjay Kumar, Setu Sinha, Varsha Singh, Ramesh Verma, Avijit Hazra, Raghunath Misra, Divya Mehrotra, Sashi Bhusan Biswal, Ankita Panigrahy, Kusum Lata Gaur, Jai Prakash Pankaj, Dharmesh Kumar Sharma, Kondeti Madhavi, Pulaganti Madhusudana, K. Narayanasamy, A. Chitra, Gajanan D. Velhal, Amit S. Bhondve, Sharminder Kaur, Shankar Prinja.

**Resources:** Maninder Pal Singh, Kavitha Rajsekar, Oshima Sachin, Jyotsna Naik, Sanjay Kumar, Setu Sinha, Varsha Singh, Prakash Patel, Ramesh Verma, Avijit Hazra, Raghunath Misra, Divya Mehrotra, Sashi Bhusan Biswal, Ankita Panigrahy, Kusum Lata Gaur, Jai Prakash Pankaj, Dharmesh Kumar Sharma, Kondeti Madhavi, Pulaganti Madhusudana, K. Narayanasamy, A. Chitra, Gajanan D. Velhal, Amit S. Bhondve, Rakesh Bahl, Sharminder Kaur, Shankar Prinja.

**Software:** Maninder Pal Singh, Riya Popli, Prakash Patel, Rakesh Bahl, Shankar Prinja.

**Supervision:** Maninder Pal Singh, Kavitha Rajsekar, Oshima Sachin, Jyotsna Naik, Shankar Prinja.

**Validation:** Maninder Pal Singh, Riya Popli, Kavitha Rajsekar, Oshima Sachin, Jyotsna Naik, Ramesh Verma, Shankar Prinja.

**Visualization:** Maninder Pal Singh, Riya Popli, Shankar Prinja.

**Writing – original draft:** Maninder Pal Singh, Shankar Prinja.

**Writing – review & editing:** Maninder Pal Singh, Riya Popli, Sehr Brar, Kavitha Rajsekar, Oshima Sachin, Jyotsna Naik, Sanjay Kumar, Setu Sinha, Varsha Singh, Prakash Patel, Ramesh Verma, Avijit Hazra, Raghunath Misra, Divya Mehrotra, Sashi Bhusan Biswal, Ankita Panigrahy, Kusum Lata Gaur, Jai Prakash Pankaj, Dharmesh Kumar Sharma, Kondeti Madhavi, Pulaganti Madhusudana, K. Narayanasamy, A. Chitra, Gajanan D. Velhal, Amit S. Bhondve, Rakesh Bahl, Sharminder Kaur, Shankar Prinja.

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
