## [Decision Letter · Decision Letter 0]

16 May 2022

PONE-D-22-03972Collecting cost data in private hospitals – Learnings from a process evaluation of CHSI study in IndiaPLOS ONE

Dear Dr. Prinja,

Thank you for submitting your manuscript to PLOS ONE. After careful consideration, we feel that it has merit but does not fully meet PLOS ONE’s publication criteria as it currently stands. Therefore, we invite you to submit a revised version of the manuscript that addresses the points raised during the review process.

Please see the comments from both the reviewers and address them in the revised manuscript. 

We look forward to receiving your revised manuscript.

Kind regards,

Alok Ranjan

Academic Editor

PLOS ONE

Journal Requirements:

We would like to acknowledge the contribution of the funding agency, the Department of Health Research (DHR), for providing the necessary funding opportunity for the ‘Costing of Health Services in India’ study. We gratefully acknowledge the efforts of state data collection teams and staff of the private hospitals in sharing the information as part of the Costing of Health Services (CHSI) study.

NO

Reviewers' comments:

Reviewer's Responses to Questions

**Comments to the Author**

1. Is the manuscript technically sound, and do the data support the conclusions?

Reviewer #1: No

Reviewer #2: Yes

2. Has the statistical analysis been performed appropriately and rigorously? 

Reviewer #1: Yes

Reviewer #2: Yes

3. Have the authors made all data underlying the findings in their manuscript fully available?

Reviewer #1: Yes

Reviewer #2: Yes

4. Is the manuscript presented in an intelligible fashion and written in standard English?

Reviewer #1: Yes

Reviewer #2: Yes

5. Review Comments to the Author

Reviewer #1: The study is on an interesting and relevant topic. However, the methods, results and discussion leave much to imagination. The write up needs to be organized better. The paper has potential but it needs improvements in many aspects. The specific queries, comments and suggestions are:

Abstract: The first line of abstract seems irrelevant to topic at hand. Also, the first two sentences in the introduction do not help in bringing attention on the real topic at hand. So does the first sentence in Discussion.

The results part in the abstract provides mean no. of months for collecting data. For how many investigators? Working full time? Please make it more specific.

Methods:

An abbreviation MRU has been used in several places but the full form has not been provided anywhere.

“Mix of top down and bottom up costing was used for CHSI” – this is very vague. Please specify.

How did the CHSI study gather data on time spent by human resources on different activities? The method of gathering data on HR’s time-use has implications for the time it will take to collect data on them. Was it collected for each category of staff? Or was it collected for each individual person on the staff? If any forms/templates were used under CHSI to collect this data, please share as annexures.

Since this is a process evaluation of CHSI (the part on private hospitals) study, it is important to provide details of methods used for data collection under CHSI and to relate the findings of this process evaluation to them.

Why have the authors decided to use Likert scale? It does not seem to add much value. Why not use qualitative interviews to find out the aspects which posed greater difficulty?

What is supplementary file S1? The text does not specify it clearly. Is it the online survey form? It seems too long to be suitable for online survey. A structured questionnaire is likely to give more accurate results rather than a lengthy semi-structured one. It asks qualitative questions like – describe challenges faced, innovative methods used to solve problems etc. An online questionnaire may not be suitable for such qualitative data. There is a question on “how the rental price was determined?” Did CHSI leave it to teams to decide how they collect this information or did it provide them specific guidance on collecting such data?

The third method was review of documented monitoring data of CHSI. Conducting in-depth interview of central monitoring team constitutes a separate method and should not be confused with “review of monitoring data”.

Sampling: Data was collected from 13 for-profit private hospitals (in country as diverse as India with a very diverse for-profit private sector). But the authors have claimed that it was nationally representative (Page 13, line 99). Was it representative? Was it randomized? How was the sample size decided? Was the sample size enough? The inter-quartile ranges reported in Table 1 suggest otherwise. Please explain the sampling procedure in detail.

Results:

The variable on months spent for data collection is vague. We don’t know whether the teams worked continuously and full time (days per month, hours per day) or part time.

The information presented in Table 2- Which part of the process evaluation provided it (out of the 3 methods – online survey, in-depth interviews, review of monitoring data?

For each result presented, please specify the method used for data collection.

The suggestions given in last column of Table 2: How are they part of results? Some of the challenges listed are unclear. What do the authors mean by “financially sensitive” or “data sharing hesitancy”. For collecting cost data on consumables, why there was no interaction with an accountant/procurement manager?

Discussion:

It is not clear what the approach was for deciding the price for a particular service? HBP of PMJAY seems to have the same price irrespective of hospital being public or private. Where private hospitals showed different cost than the public ones, which one was used to decide HBP rate?

What about the challenge of hospitals giving exaggerated figures of costs involved? Was it encountered? How was it dealt with?

The main theme emerging is that private hospitals were reluctant to share data. Even so, when the data was needed for a government study. Why is there such reluctance in private sector? The discussion should explore this.

What is the experience available from other countries regarding difficulties in collection of cost data? Any mention of international literature is absent in the current manuscript. This gap should be addressed. The introduction should include literature review. The discussion should compare the main findings with other studies.

In the section on data collection in public hospitals versus private hospitals, where is the information regarding data collection in public hospitals coming from. Is it from another reference or collected through the online survey in current process evaluation? Please specify. It is useful to compare the challenges involved in public versus private hospitals. But the data sources need to be clear. The discussion on this comparison should go beyond comparing the numbers and delve into nature of challenges involved and underlying reasons.

What was the comparison (of data collect difficulties) between for-profit versus non-profit hospitals? The current study covered both kinds of hospitals. It will be very important to compare these two categories in results and discussion. It is shocking that no comparison has been provided. Clubbing the two very different categories of non-government hospitals throughout the paper does not help.

The recommendation given is to make such information sharing compulsory under PMJAY empanelment contracts. What is the experience on private sector adhering to contracts under such insurance schemes in India? Why not strengthen the existing regulatory mechanisms to improve transparency? Private hospitals are registered as clinical establishments in India. What are the reporting requirements for them? Are they also registered as commercial businesses? What kind of reporting standards are applicable in India?

Reviewer #2: It is an important and difficult study to dive in private sector. It is important to mention that it is a good effort. However, manuscript needs some additions for better clarity.

Please see if you can rework the title as “Challenges and solutions of inclusion of private hospitals in Costing of services” OR “Situational analysis for inclusion of private hospitals in Costing of Health Services in India”

Introduction

1. Include state of private hospitals in India, like number of hospitals. Categorization of hospitals (large and small) would be useful.

Methods:

1. Mention inclusion criteria of hospitals for the study.

2. Mention total number of private hospitals in each included state.

3. Mention refusal rate (overall and in each state), if not state the potential reasons of non-refusal.

Discussion:

Introduce limitation section in discussion section focusing;

1. Discuss about potential selection and observer bias in present study.

2. Whether results are generalizable or not?

If possible, a small review of legal challenges (if any) can be included to assess potential conflict if data sharing made mandatory by private hospitals.

6. PLOS authors have the option to publish the peer review history of their article (what does this mean?). If published, this will include your full peer review and any attached files.

Reviewer #1: No

Reviewer #2: No

---

## [Author Response · Author response to Decision Letter 0]

9 Aug 2022

Authors Response to Editor and Reviewer Comments

First of all, we would like to thank the reviewers and the Editorial team of the PlosOne for giving us this opportunity for revising the manuscript. We appreciate the useful comments which have helped improve the manuscript. The point-wise responses to reviewer comments are listed below. 

Reviewer#1 

1. Abstract: The first line of abstract seems irrelevant to topic at hand. Also, the first two sentences in the introduction do not help in bringing attention on the real topic at hand. So does the first sentence in Discussion.

Author’s reply: We thank the reviewer for this valuable comment. We have introduced the term strategic purchasing to highlight the role of private sector in provisioning of care under the publicly financed health insurance schemes such as PM-JAY. We have revised the abstract, introduction and discussion to make it more clear. In abstract (P3 line 35-37), ‘Ayushman Bharat Pradhan Mantri Jan Aarogya Yojana (AB PM-JAY) has enabled the Government of India to become a strategic purchaser of health care services from private providers.’ 

In introduction (P4 line 66-69), ‘In India, the government launched the flagship program, Ayushman Bharat (AB) in 2018 to provide universal health coverage (UHC). It is a two-pronged strategy providing both primary care through Health & Wellness Centres (AB HWCs), and secondary and tertiary care services through the Pradhan Mantri Jan Arogya Yojana (AB PM-JAY).(1)’

In discussion (P19 line 349-351), ‘In the PM-JAY scheme, 56% of the empanelled health care providers (EHCPs) are private hospitals. Private hospitals are catering to 63% of the PM-JAY claims volume, and 75% in terms of claims amount.(16)’

2. Abstract: The results part in the abstract provides mean no. of months for collecting data. For how many investigators? Working full time? Please make it more specific.

Author’s reply: We thank the reviewer for this comment. We have revised the abstract and provided data in terms of person-days (P 3 line 48-50), ‘The mean time taken for data collection in a private hospital was 9.31 (� 1.0) person months including time for obtaining permissions, actual data collection and entry, and addressing queries for data completeness and quality.’ Further, the details of the data collection team was added in the manuscript (P10 line 250-253), ‘Each state data collection team had 3 project staff (2 field officers and 1 administration assistant) guided by 1 to 3 state co-investigators. The field data collection teams comprising of 3 full-time staff which worked for 40 hours a week except for Sundays and public holidays of the respective state.’

3. Methods: An abbreviation MRU has been used in several places but the full form has not been provided anywhere.

Author’s reply: The term MRU is used for the multi-disciplinary unit set up by the Department of Health Research which collect the data as part of the CHSI study. For consistency we have replaced the abbreviation MRU with ‘field study team’ in the manuscript.

4. “Mix of top-down and bottom-up costing was used for CHSI” – this is very vague. Please specify.

Author’s reply: We had earlier provided a reference to the main protocol paper of this study. However, in view of the reviewer comment, we have now added more details of costing methodology to the manuscript (P7 146-155), ‘The study uses standard principles of economic costing and the micro-costing approach.(17, 23) A mixed micro-costing approach (top-down and bottom-up) from a payer’s perspective was used to estimate health system cost. While a pure bottom-up costing methodology may provide the most detailed and granular estimation of cost, lack of disaggregated data and electronic data management systems makes its execution difficult for certain resources. Hence, for resources such as utilities, maintenance, and in certain cases the drugs and consumables, The top-down methodology was used. However, our top-down method provided much more granular level of aggregation at the level of a cost centre, rather than entire health facility. (24) Such a methodology is commonly applied in the costing studies carried out in India and other LMIC settings.(17, 25)’

5. How did the CHSI study gather data on time spent by human resources on different activities? The method of gathering data on HR’s time-use has implications for the time it will take to collect data on them. Was it collected for each category of staff? Or was it collected for each individual person on the staff? If any forms/templates were used under CHSI to collect this data, please share as annexures.

Author’s reply: We used time allocation interviews to collect data on human resources. We have added the same in the manuscript (P7 line 159-165), ‘Within each cost centre human resources accounted for highest share in the cost of service delivery. As, human resources are shared between different cost centres, time allocation interviews were conducted to understand the work pattern. The time allocation interviews were undertaken for each category of the staff such as specialist, doctors, nurses, technicians to name a few. Within each staff category at least 25% of the staff was interviewed.(17)’ We have added the time allocation data collection tool in the supplementary material S2 

Supplementary S2: Costing of Health Services (CHSI) Time Allocation Tool

Staff Member Code (Enter Code as entered in Section 1): ............................................

Service code no Activity name Type of activity Fixed schedule activity Routine activity

 Fixed schedule Routine Frequency (once in a week/once in month/twice a week etc.) * Hours per day of activity Days for which the activity was done during the reference year Time per person (in minutes) (a) Number of beneficiaries on a routine day (b) If not (a) and (b) then how much time to do the activity

1. Outpatient (OP) department 

2. Inpatient (IP) department 

3. Intensive care unit (ICU) 

4. Operation Theatre (OT) 

5. General Administration 

6. Teaching/Training 

7. Workshop/Conference 

8. Outreach 

9. Meetings 

10. Research 

11. Others (Specify) 

• Codes: ‘1’ for once a year participation, ‘2’ for twice a year, 3 for thrice a year participation, 4 for quarterly participation, 5 for once every two 

• months, 6 for monthly participation, 7 for fortnightly participation, 8 for weekly participation, 9 for twice a week participation, 10 for thrice

• a week participation.

6. Since this is a process evaluation of CHSI (the part on private hospitals) study, it is important to provide details of methods used for data collection under CHSI and to relate the findings of this process evaluation to them.

Author’s reply: We thank the reviewer for the comment. We have included information on the methodology of the CHSI study in the revised manuscript. The findings from process evaluation have been related to the methodological aspects. We have discussed the results in terms share of overall cost incurred by individual input resources and the amount of data collection time spent with respect to the share of input resources in the overall cost (P14 line 297-303), ‘Further, a comparison was undertaken between the proportion of time taken for data collection of each input resource and their contribution to the total cost. It was observed that human resources contribute the maximum in total cost (38%), as well as in time taken for data collection (28%). The capital/building and consumables & drugs contribute 44% to the total cost but only 18% of the time was spent to collect data for these inputs. On the other hand, data on furniture and equipment consumed 34% of the time but contributes only 7% to the total cost. (Figure 3)’ 

7. Why have the authors decided to use Likert scale? It does not seem to add much value. Why not use qualitative interviews to find out the aspects which posed greater difficulty?

Author’s reply: We thank the reviewer for the comment. As part of the cost data collection, each field team had a different set of challenges and difficulties. Therefore, to quantify it across different states we used a Likert scale. Further, we used the same scale in our published article on process evaluation in the public tertiary hospitals in PlosOne. Therefore, it will be easier for the readers to compare the findings which will help them to plan the future costing studies. 

8. What is supplementary file S1? The text does not specify it clearly. Is it the online survey form? It seems too long to be suitable for online survey. A structured questionnaire is likely to give more accurate results rather than a lengthy semi-structured one. It asks qualitative questions like – describe challenges faced, innovative methods used to solve problems etc. An online questionnaire may not be suitable for such qualitative data. There is a question on “how the rental price was determined?” Did CHSI leave it to teams to decide how they collect this information or did it provide them specific guidance on collecting such data?

Author’s reply: We thank the reviewer for the comment. We have used multiple data collection tools i.e. an online survey using a semi-structured questionnaire, in-depth interviews, and a review of monitoring data and interviews with the central analysis tool. The majority of the questions in the online tool were structured and for the qualitative information we have used the in-depth interviews.. Further we have added the online tool in the supplementary file S3 which was pilot tested and used for data collection from the public tertiary hospitals. The results of which are already published in the PlosOne. (https://doi.org/10.1371/journal.pone.0232873) Further, the field data collections were trained during in-person training (during public hospital data collection before covid pandemic) and online training before starting the data collection in the private hospitals. 

Further, as part of the CHSI study the field teams were trained through multiple in-person and online trainings on cost data collection over the period of 3.5 years. For each input resources, how the data should be collected was explained to each team and a guidelines was shared with each team. Further, at the start of the CHSI study we conducted a training of trainers (ToT) where state co-investigators and field staff was provided hands on training for cost data collection in the healthcare facilities.

9. The third method was review of documented monitoring data of CHSI. Conducting in-depth interview of central monitoring team constitutes a separate method and should not be confused with “review of monitoring data”.

Author’s reply: We thank the reviewer for the comment. To avoid the confusion in the terminology we have revised the heading to ‘Review of monitoring data and interviews with central team’ and revised the manuscript (P9 line 222-227), ‘The CHSI central data analysis team monitors the state-level data collection and maintains the data on time taken to complete data collection, need for on-the-job support in collecting data, quality of data etc. This data maintained by the central analysis team was used to calculate the time spent on data collection by each field study team. The central team was interviewed to understand the process of quality assurance.’

10. Sampling: Data was collected from 13 for-profit private hospitals (in country as diverse as India with a very diverse for-profit private sector). But the authors have claimed that it was nationally representative (Page 13, line 99). Was it representative? Was it randomized? How was the sample size decided? Was the sample size enough? The inter-quartile ranges reported in Table 1 suggest otherwise. Please explain the sampling procedure in detail.

Author’s reply: The CHSI study is a national level costing study commissioned by the Indian government’s Department of Health Research. This study covers a sample of 11 public sector tertiary hospitals, 27 public sector district hospitals and 38 private hospitals. We do recognize the heterogeneity in the cost data and the influence of various supply-side and demand-side factors on cost of providing care. The sampling of the hospitals was carefully designed to ensure that the estimates are as generalizable as possible. A multistage stratified sampling method was used for the selection of states. The states were selected to represent the heterogeneity based on geography, health indicators, net state domestic product (NSDP) and health workforce density. The states included in the study sample were Jammu and Kashmir, Rajasthan, Uttar Pradesh, Bihar, West Bengal, Gujarat, Odisha, Maharashtra, Andhra Pradesh, and Tamil Nadu. Further a stratification plan to select districts, and hospitals has also been described in the revised manuscript. In the earlier draft of the manuscript, these methodological details were not included as the paper specifically focuses on the process evaluation of data collection in private hospitals. However, considering the reviewer comment, we have added details of methodology as well as added a new supplementary appendix to explain the same.

We have revised the manuscript to add the sampling strategy for the private hospitals (P6 line138-142 and P7 line 144-145), ‘A stratified sampling methodology was used for selection of private hospitals. The 75% of the sampled hospitals were for-profit and 25% were not for-profit hospitals. This was based on the proportion of private hospitals in each of the category.(18) Further, the location of hospital (i.e. classification of cities used by the government in India tier 1, tier 2 and tier 3) and number of hospitalisation beds (up to 50 bedded, 50-200 bedded and more than 200 bedded) was used as a sampling criteria. No single speciality hospital was included in the study. From each state a sample of 3-4 hospitals was included in the study.’

Further, we have added the broad sampling strategy for the CHSI study in the supplementary material S1 to explain the sampling methodology for the CHSI study. 

‘Supplementary S1: Costing of Health Services (CHSI) Study Sampling

A multistage stratified sampling method was used. The states were selected to represent the heterogeneity based on geography, health indicators, net state domestic product (NSDP) and health workforce density. The state specific factors are i.e. population, geography, Sustainable Development Goals India Index, Human Development Index, per capita NSDP and health workforce density (per 10,000 population) are shown in the table below for the sampled states.

Table 1: Costing of Health Services (CHSI) Sampling Framework

State Population (Census 2011) Geography SDG India Index 2018* 

(0-100) Human Development Index 

(HDI) Gross State Domestic Product# 

(GSDP) Health workforce Density

Jammu & Kashmir 12,541,302 North 53 Medium Low Medium

New Delhi 18,345,784 North 62 High High High

Rajasthan 68,548,437 North 59 Low Low Low

Uttar Pradesh 199,812,341 North 48 Low Low Low

Gujarat 60,439,692 West 64 Medium Medium Medium

Maharashtra 112,374,333 West 64 Medium High High

Andhra Pradesh 84,580,777 South 64 Medium Medium Medium

Tamil Nadu 72,147,030 South 66 High High High

Bihar 104,099,452 East 42 Low Low Low

West Bengal 91,276,115 East 56 Low Low High

Odisha 41,974,218 East 51 Low Low Low

Meghalaya 2,966,889 North-East 52 Medium Low Low

*SDG: Sustainable Development Goals; Source: NITI Aayog 2018, #Reserve Bank of India

Source: Prinja S, Singh MP, Guinness L, Rajsekar K, Bhargava B. Establishing reference costs for the health benefit packages under universal health coverage in India: cost of health services in India (CHSI) protocol. BMJ Open. 2020;10(7):e035170.

Within each state, a tertiary level medical institution was chosen. At the secondary care level, costing was done at three district hospitals in each state. The districts were randomly selected from each of the three tertiles of the district composite development score ranking. This index was based on an aggregation of socioeconomic, demographic and health service usage indicators.’

11. Results: The variable on months spent for data collection is vague. We don’t know whether the teams worked continuously and full time (days per month, hours per day) or part time.

Author’s reply: We thank the reviewer for this comment. The time for data collection is for a team of three full time staff. We have revised the manuscript (P10 line 251-253), ‘The field data collection teams comprising of 3 full-time staff which worked for 40 hours a week except for Sundays and public holidays of the respective state.’ and (P11 line 272-276) , ‘ The mean time taken for overall data collection per private hospital for a field data collection team comprising of three full-time staff, which includes obtaining permissions, actual data collection and entry, queries and final submission, was 3.94 months respectively. The time of data collection ranged from 1.21 months to 6.76 months. (Figure 1) On an average, data collection in one hospital covering all its services required 9.31 (� 1.0) person months.’

12. The information presented in Table 2- Which part of the process evaluation provided it (out of the 3 methods – online survey, in-depth interviews, review of monitoring data?

Author’s reply: We have revised the manuscript to add that the data presented in the table 2 is derived from the online survey and in-depth interviews (P15 line 325-326), ‘The data presented in the table 2 was derived from the online survey and in-depth interviews.’

13. For each result presented, please specify the method used for data collection.

Author’s reply: We have revised the manuscript to specify the methods used for each of the result generated. 

(P11 line 259-261), ‘The data on time period for primary data collection, hospitals approached and refusal rate was calculated from the online survey and review of monitoring data and interviews with central team.’ 

(P12 line 294-295), ‘The share of time required for each type of input resources was calculated from the online survey.’ 

(P14 line 306-308), ‘The specific challenges and rating of difficulty in collecting data on each input resource are described in this section which are derived from the online survey and in-depth interviews with the state field teams. (Figure 4)’

(P15 line 325-326), ‘An overview of data collection for different input resources was provided in table 2. The data presented in the table 2 was derived from the online survey and in-depth interviews.’

14. The suggestions given in last column of Table 2: How are they part of results? Some of the challenges listed are unclear. What do the authors mean by “financially sensitive” or “data sharing hesitancy”. For collecting cost data on consumables, why there was no interaction with an accountant/procurement manager?

Author’s reply: We thank the reviewer for the comment. The suggestions given in the last column were generated based on the in-depth interviews and online survey and hence included in the results section. We have added same in the manuscript (P15 line 325-326), ‘An overview of data collection for different input resources was provided in table 2. The data presented in the table 2 was derived from the online survey and in-depth interviews.’ 

We have used the term ‘financially sensitive’ information which means that the salaries and procurement prices are based on the price negotiations. These negotiations take place between the human resource managers of hospitals and future employees for salary, as well as between the hospital administrators and the manufacturers/ sellers for various drugs and consumables. The salary data is also sensitive in view of tax liabilities. Further, the prices of drugs and consumables are sensitive in view of the price confidentiality with the supplier, and also since the hospitals do not want to reflect on hospital profits. We have added an operation definition of the term financially sensitive in the footnote of the table 2 (P18 table 2), ‘*The term financially sensitive means that the information pertaining to salaries and procurement prices are based on internal mechanisms and negotiations. There disclosure may influence the future negotiations for the private hospitals.’

Regarding the consumables, as the costing required the disaggregated data for each cost centre was required. Therefore, pharmacist and staff nurse in charge was approached first. In some hospitals the ‘person in charge’ was involved who may be accountant or procurement officer or any other designation.

15. Discussion: It is not clear what the approach was for deciding the price for a particular service? HBP of PMJAY seems to have the same price irrespective of hospital being public or private. Where private hospitals showed different cost than the public ones, which one was used to decide HBP rate?

Author’s reply: Under the PMJAY scheme, the reimbursement prices are continuously revised since the start of the scheme. Three revisions in HBP prices have taken place in 2019, 2021 and 2022. The prices during each of the previous revision was based on the cost of delivery of healthcare services based on the tertiary public hospitals as cost evidence from private hospitals was not available. More recently, a differential pricing based on the tier-wise classification of location of city has been introduced. This differential pricing has taken into consideration the cost in both public and private hospitals. We have revised the manuscript (P19 line 353-357), ‘Currently, public and private hospitals are paid uniformly under the PMJAY scheme. Under the latest PM-JAY price revision in the 2022, the National Health Authority (NHA) introduced differential pricing based on the tier classification of the cities. This has made use of cost of services in both the public and the private hospitals.(30)’

16. What about the challenge of hospitals giving exaggerated figures of costs involved? Was it encountered? How was it dealt with?

Author’s reply: We thank the reviewer for this important question. We were careful to ensure that the private hospitals neither present exaggerated figures, nor under-report the prices of resources in our data collection. The chances that private hospitals may under-report the figures considering tax and registration compliance, was perceived higher. We used several triangulation methods by comparing the data provided with similar unit prices obtained from other hospitals in the same city, or state, and with the price data from the National Health System Cost Database to ensure that there is no significant range and consistency deviation (P22 line 464-468), ‘Further, due consideration should be given for the data quality checks so that the reported cost data is not exaggerated or under-reported such as using several triangulation methods by comparing the data provided with similar unit prices obtained from other hospitals in the same city, or state, and with the price data from the National Health System Cost Database of India.(32)’ 

17. The main theme emerging is that private hospitals were reluctant to share data. Even so, when the data was needed for a government study. Why is there such reluctance in private sector? The discussion should explore this.

Author’s reply: We thank the reviewer for this comment. The cost data collection requires quantity and prices of all the input resources which is very time-consuming in absences of standardized data management systems. Based on our experience from CHSI study and previous costing studies, the study respondent (from private hospital) perceives the data collection may lead to some audit, which may reflect badly for them or institution. Moreover, since the prices for drugs, consumables and equipment has confidentiality issue related to the hospital profit margins. Similarly, the salary data has linked liability of taxes, which may not be revealed to avoid paying tax. Hence, there is reluctance to share the data, more commonly in private hospitals. In public hospitals, the reluctance to share data on quantity of resources consumed can only arise if the hospital records are not well maintained and there can be a mismatch between the hospital records and its reports submitted to higher facility. The data on time allocation may be influenced by the respondent’s social desirability bias to be seen an performing in an ideal manner. We have highlighted it in the discussion section (P19 376-379), ‘In low-and middle-income countries (LMICs), cost data collection has been reported to have numerous challenges such as willingness to participate, sharing of data, the form of data management systems, lack of electronic records, lack of disaggregated data, multiple sources to name a few.(22, 28-30)’ and (P20 line 390-396 and P21 397-402), ‘The data collection in private hospitals pertaining to human resources in terms of salaries, leaves and time allocation was reported as most difficult and time-consuming by field investigating teams in terms of initially seeking permission, as well as actual data collection time. Based on our experience from CHSI study and previous costing studies, the study respondent (from private hospital) perceives the data collection may lead to some audit, which may reflect badly for them or institution. Moreover, since the prices for drugs, consumables and equipment has confidentiality issue related to the hospital profit margins. Similarly, the salary data has linked liability of taxes, which may not be revealed to avoid paying tax. Hence, there is reluctance to share the data, more commonly in private hospitals. In public hospitals, the reluctance to share data on quantity of resources consumed can only arise if the hospital records are not well maintained and there can be a mismatch between the hospital records and its reports submitted to higher facility. The data on time allocation may be influenced by the respondent’s social desirability bias to be seen an performing in an ideal manner.’

18. What is the experience available from other countries regarding difficulties in collection of cost data? Any mention of international literature is absent in the current manuscript. This gap should be addressed. The introduction should include literature review. The discussion should compare the main findings with other studies.

Author’s reply: We have added a section on challenges of cost data collection in the low-and middle-income countries (LMICs) in the introduction section. However, to the best of our knowledge the challenges and solutions are provided in a generic way but not specifically in terms of each input resource and capturing the time for data collection. The published literature focuses on the advantages and disadvantages of different methodological approaches (top-down vs bottom-up, financial vs economic, normative vs real-world) along with their feasibility in terms of availability of disaggregated data. On the contrary, the availability of patient level resource use data in developed countries using electronic health records simplifies the whole process with use of software packages. We have revised the manuscript (P5 105-111), ‘The health system costing studies require a complex set of information for different input resources which is a labour intensive and time consuming. The process on data collection is contingent upon the costing approach (financial or economic), perspective (health system or patient), methods (top-down or bottom-up or mixed), willingness to share data, availability of data management systems to name a few. (21, 22) These challenges are further exacerbated in the low-and middle-income countries (LMICs) such as India due to insufficient data management systems and requirement of data extraction from physical registers.(23-25)’

19. In the section on data collection in public hospitals versus private hospitals, where is the information regarding data collection in public hospitals coming from. Is it from another reference or collected through the online survey in current process evaluation? Please specify. It is useful to compare the challenges involved in public versus private hospitals. But the data sources need to be clear. The discussion on this comparison should go beyond comparing the numbers and delve into nature of challenges involved and underlying reasons.

Author’s reply: Online survey tool was used for both public and private hospitals we have revised the manuscript (P20 381-382), ‘Online survey tool, in-depth interviews and review of monitoring data and interviews with central team was used to collect process evaluation data for both tertiary public and private hospitals.(24)’ We have highlighted the difference in data management systems in the public and private hospitals as well as key personnel responsible for sharing data with the field teams (P20 line 412-420 & P21 line 425), ‘The actual data collection time (person-days) for time allocation interviews is almost double in private hospitals as reported for public tertiary hospitals. The level of difficulty to access the data for equipment, furniture and consumables were higher in public sector hospitals as compared to the private hospitals. One of the reasons can be the difference in data management systems. In private hospitals, computer-generated reports of data were made available to the teams however in public tertiary hospitals data was extracted from physical registers. Another reason could be that multiple individuals were responsible for management within the same cost centre in the public sector hospitals, which increases the extent of permissions, need for explaining the purpose of data collection, and hence the overall data collection time. However, in the private hospital's data, extraction was dependent on quite a few staff members.’

20. What was the comparison (of data collect difficulties) between for-profit versus non-profit hospitals? The current study covered both kinds of hospitals. It will be very important to compare these two categories in results and discussion. It is shocking that no comparison has been provided. Clubbing the two very different categories of non-government hospitals throughout the paper does not help.

Author’s reply: We thank the reviewer for this comment. We agree with the reviewer that the process of registration is different for the for-profit and not for-profit hospitals in India. However, the data collection approaches and methods remain the same. The difference is likely to be more pronounced in terms of unit cost estimates between for for-profit and not-for-profit hospitals. In our study, we found no significant differences in terms of challenges, negotiation time, cooperation etc between the two different types of hospitals. 

21. The recommendation given is to make such information sharing compulsory under PMJAY empanelment contracts. What is the experience on private sector adhering to contracts under such insurance schemes in India? Why not strengthen the existing regulatory mechanisms to improve transparency? Private hospitals are registered as clinical establishments in India. What are the reporting requirements for them? Are they also registered as commercial businesses? What kind of reporting standards are applicable in India?

Author’s reply: We thank the reviewer for the comment. A Central legislation — Clinical Establishments (Registration and Regulation) Act (CEA)— was put in place in 2010. The new law was applicable to all systems of medicine and all public and private medical establishments. However, in recent working paper entitled, ‘.Analysing Regulation of Private Healthcare in India With focus on Clinical Establishments Acts Current status, challenges and recommendations’ highlighted the slow implementation of the CEA. The provision of cost data should be part of empanelment requirement under the PM-JAY as it covers a part of the full universe of the private sector hospitals in India. Hence, the CEA should be strengthened and gradually the reporting of cost of health services should be integrated and strengthened as part of the provisions of the CEA. We have added a section on it in the limitation section of the manuscript (P21 438-443), ‘Secondly, the Clinical Establishments (Registration and Regulation) Act (CEA) was enacted by the central government in 2010.(32) The CEA is applicable to all systems of medicine and all the public and private hospitals. However, the implementation of CEA remains slow. Therefore, the reporting of mandatory cost data should be part of the empanelment requirement under the PM-JAY. Further, the implementation of CEA should be expedited and gradually reporting of cost of health services should be integrated to the provisions of the CEA.’

 

Reviewer#2

1. Please see if you can rework the title as “Challenges and solutions of inclusion of private hospitals in Costing of services” OR “Situational analysis for inclusion of private hospitals in Costing of Health Services in India”

Author’s reply: We thank the reviewer for this comment. As suggested by the reviewer, we have reframed the title of the current paper as ‘Process evaluation of CHSI Costing study – Challenges and solutions for cost data collection in private hospitals in India.’ 

2. Introduction - Include state of private hospitals in India, like number of hospitals. Categorization of hospitals (large and small) would be useful. 

Author’s reply: We thank the reviewer for this comment. We have added a brief overview of private hospitals in India in the introduction section (P 5 line 94-99), ‘Private sector hospitals in India have grown and diversified in terms of service provisioning, infrastructure, ownership to name a few.(18) There is no centralized registration of private hospitals in India. Private sector comprises for-profit and not for-profit hospitals ranging from large corporate hospitals, super-speciality facilities to medium and small nursing homes and even single doctor dispensaries.(19) In India, there are 43,486 private hospitals accounting for 62% of the health infrastructure of the country.(20)’

3. Methods:

1. Mention inclusion criteria of hospitals for the study. 

2. Mention total number of private hospitals in each included state. 

3. Mention refusal rate (overall and in each state), if not state the potential reasons of non-refusal. 

Author’s reply: We have added the sampling strategy, inclusion criteria and hospitals sampled per state in the methods section (P6 line 134-137 and P7 line 139-140), ‘A stratified sampling methodology was used for selection of the hospitals, wherein 75% of the sampled hospitals were for-profit and 25% were not for-profit hospitals. This was based on the proportion of private hospitals in each of the category.(18) Further, the location of hospital (i.e. classification of cities used by the government in India tier 1, tier 2 and tier 3) and number of hospitalisation beds (up to 50 bedded, 50-200 bedded and more than 200 bedded) was used as a sampling criteria. No single speciality hospital was included in the study. From each state a sample of 3-4 hospitals was included in the study.’

Further, the refusal rate is added in the results section (P11 line 258-264), ‘In each state on an average 6 hospitals were approached to participate in the CHSI study. A minimum of 3 hospitals were approached in Odisha and Gujarat, and there was zero refusal rate. In Uttar Pradesh 7 hospitals were approached and 5 (71%) refused to participate in the study. In states of Rajasthan and West Bengal 9 hospitals were approached and there was a refusal rate of 67% in both the states. In Andhra Pradesh 6 hospitals were approached and 3 (50%) refused to consent for the study. The overall refusal rate among private hospitals across the states was 45%.’

4. Discussion: Introduce limitation section in discussion section focusing; 

1. Discuss about potential selection and observer bias in present study. 

2. Whether results are generalizable or not? 

If possible, a small review of legal challenges (if any) can be included to assess potential conflict if data sharing made mandatory by private hospitals. 

Author’s reply: We thank the reviewer for this comment. We have added a limitation section to address the queries related to selection bias, generalizability and legal challenges (P21 line 406-412 and P22 line 413-418), ‘Our study has a few limitations. Firstly, while the overall sampling plan was adhered to, there was a tendency of field data collection teams to choose the private hospitals where they had personal contacts with the owner/s or management of the hospital to facilitate data collection. It may lead to a selection bias in sampling of the private hospitals. Secondly, registration of for-profit and not for-profit is under different laws in India. Further, few states have enacted the clinical establishment act which may also vary. Therefore, for the mandatory data sharing from private hospitals under the PM-JAY should give a due consideration to that the provisions are not contradictory to the existing legal framework. Thirdly, due to lack of centralized registration mechanisms for private hospitals and considering vast heterogeneity in organisation of private hospitals, the result of the present study may not be generalizable at the national level. However, the results will provide an importance guidance for the planning and execution of future rounds of cost data collection in the private hospitals.’

---

## [Decision Letter · Decision Letter 1]

6 Oct 2022

CHSI costing study – Challenges and solutions for cost data collection in private hospitals in India

PONE-D-22-03972R1

Dear Dr. Prinja,

We’re pleased to inform you that your manuscript has been judged scientifically suitable for publication and will be formally accepted for publication once it meets all outstanding technical requirements.

Kind regards,

Alok Ranjan

Academic Editor

PLOS ONE

Additional Editor Comments (optional):

Reviewers' comments:

Reviewer's Responses to Questions

**Comments to the Author**

1. If the authors have adequately addressed your comments raised in a previous round of review and you feel that this manuscript is now acceptable for publication, you may indicate that here to bypass the “Comments to the Author” section, enter your conflict of interest statement in the “Confidential to Editor” section, and submit your "Accept" recommendation.

Reviewer #1: All comments have been addressed

Reviewer #2: All comments have been addressed

2. Is the manuscript technically sound, and do the data support the conclusions?

Reviewer #1: Yes

Reviewer #2: Yes

3. Has the statistical analysis been performed appropriately and rigorously? 

Reviewer #1: Yes

Reviewer #2: Yes

4. Have the authors made all data underlying the findings in their manuscript fully available?

Reviewer #1: Yes

Reviewer #2: Yes

5. Is the manuscript presented in an intelligible fashion and written in standard English?

Reviewer #1: Yes

Reviewer #2: Yes

6. Review Comments to the Author

Reviewer #1: This is an interesting study and it addresses a relevant topic. The authors have addressed the comments. I have no further comments.

Reviewer #2: All comments in the revised manuscript have been addressed. Authors have given satisfactory responses to all the comments. They have edited the manuscript at length and presented with an improvement. In my opinion, the manuscript can be accepted for publication.

7. PLOS authors have the option to publish the peer review history of their article (what does this mean?). If published, this will include your full peer review and any attached files.

Reviewer #1: **Yes: **Samir Garg

Reviewer #2: No

---

## [Editor Report · Acceptance letter]

2 Dec 2022

PONE-D-22-03972R1 

CHSI costing study – Challenges and solutions for cost data collection in private hospitals in India 

Dear Dr. Prinja:

I'm pleased to inform you that your manuscript has been deemed suitable for publication in PLOS ONE. Congratulations! Your manuscript is now with our production department. 

Kind regards, 

on behalf of

Dr. Alok Ranjan 

Academic Editor

PLOS ONE